# Adapting Techniques to Improve Efficiency in Radio Frequency Power Amplifiers for Visible Light Communications

**Daniel G. Aller** [1,*], **Diego G. Lamar** [1], **Juan Rodriguez** [2], **Pablo F. Miaja** [1],
**Valentin Francisco Romero** [3], **Jose Mendiolagoitia** [3] **and Javier Sebastian** [1]

[1]    Power Supplies Group—Electrical Engineering Department of the University of Oviedo, 33204 Gijon, Spain;
       gonzalezdiego@uniovi.es (D.G.L.); fernandezmiapablo@uniovi.es (P.F.M.); sebas@uniovi.es (J.S.)

[2]    Center for Industrial Electronics—Polytechnic University of Madrid, 28006 Madrid, Spain;
       juan.rodriguezm@upm.es

[3]    Thyssenkrupp Elevator Innovation Center S A U—Thyssenkrupp AG, 33203 Gijon, Spain;
       valentin-francisco.romero@thyssenkrupp.com (V.F.R.); jose.mendi@thyssenkrupp.com (J.M.)

*    Correspondence: garciaadaniel@uniovi.es; Tel.: +34-985-182-578

**Abstract:** It is well known that modern wireless communications systems need linear, wide bandwidth, efficient Radio Frequency Power Amplifiers (RFPAs). However, conventional configurations of RFPAs based on Class A, Class B, and Class AB exhibit extremely low efficiencies when they manage signals with a high Peak-to-Average Power Ratio (PAPR). Traditionally, a number of techniques have been proposed either to achieve linearity in the case of efficient Switching-Mode RFPAs or to improve the efficiency of linear RFPAs. There are two categories in the application of aforementioned techniques. First, techniques based on the use of Switching-Mode DC–DC converters with a very-fast-output response (faster than 1 μs). Second, techniques based on the interaction of several RFPAs. The current expansion of these techniques is mainly due to their application in cellphone networks, but they can also be applied in other promising wireless communications systems such as Visible Light Communication (VLC). The main contribution of this paper is to show how Envelope Tracking (ET), Envelope and Elimination (EER), Outphasing, and Doherty techniques can be helpful in developing more efficient VLC transmitters capable of reaching high bit-rates (higher than 1 Mbps) by using advance modulation schemes. Finally, two examples based on the application of the Outphasing technique and the use of a Linear-Assisted Envelope Amplifier (EA) to VLC are presented and experimentally verified.

**Keywords:** visible light communication (VLC); light emitting diodes (LED); switching-mode; DC–DC converters and radio frequency power amplifiers (RFPA)

## 1. Introduction

Currently, the use of wireless communication systems is essential for the present and the future of our society [1]. Furthermore, the speed demanded by each communication service is continuously growing due to the high bit-rate required (higher than 1 Mbps) by mainstream services. As a result, the Radio Frequency (RF) spectrum is already close to congestion and hence enabling the data traffic predicted for upcoming years requires further research into new wireless communication technologies.

Visible Light Communication (VLC) is one of the most promising solutions for alleviating the saturation of the RF spectrum [2–4]. VLC uses the wide, unlicensed visible light spectrum (430–750 THz range) to transmit information. The strength and potential of this approach become apparent when the communication task is merged with the lighting functionality of Light-Emitting Diode (LED) lamps.

Thus, the existing lighting infrastructure can be partially adapted to incorporate communication features. Since its introduction in 2004 [1], VLC has gained attention due to the widespread use of LEDs in lighting installations. Besides their longer lifespan, higher power efficiency, and the fact that they are more environmentally friendly than other lighting technologies, LEDs are able to change their light intensity very quickly, making this technology the most promising one in terms of enabling communication functionalities.

The design of the LED driver with communication functionalities (VLC-LED driver) is one of the main challenges of the VLC-transmitter. The VLC-LED driver is responsible for two tasks: guaranteeing the desired lighting level and reproducing, with a high degree of accuracy, the communication signal originating from the other kind circuitry of the VLC-transmitter, which processes the data.

Traditionally, the literature focuses on the most important VLC contributions in data processing, advanced modulation schemes, LED modeling and experimental verification using low power prototypes. In other words, the main efforts have been carried out to increase the spectral efficiency of VLC-LED drivers in order to achieve very high bit-rates (higher than 10 Mbps). However, a major issue with regard to VLC-LED drivers is not taken into account: the high efficiency of LED lighting is not only due to the high efficiency of the LEDs in converting electrical power into lighting power, but also to the high efficiency achieved by the VLC-LED driver that delivers the electrical power. Therefore, the VLC-LED driver must also be power efficient [5]. The VLC-LED drivers for VLC-transmitters proposed to date, which can be classified into two subsets, do not achieve both high power efficiency (higher than 90%) and high bit-rates (higher than 1 Mbps):

- VLC-LED drivers based on switching the LEDs on and off. Figure 1a shows a conventional structure based on a high efficiency, limited-output-response Switching-Mode DC–DC converter. In this example, a power MOSFET is placed in series to turn the LED on and off. Other structures with the MOSFET placed in parallel are also valid. This VLC-LED driver is simple and high-power efficient (higher than 90%). Only Pulse-Based Modulation (PBM) schemes can be reproduced using this solution [6–9]; however, PBM schemes are not the best option for providing a high bit-rate and for avoiding the multipath issue (an issue arising from the different paths that the light rays can follow from the light focus to the receiver). Although some PBM proposed solutions achieve a very high bit-rate (10 Mbps) at higher power levels associated with the light emitted for communications (up to 10 W) [8,9], they are far from the theoretically achievable bit-rate using advanced modulation schemes.
- VLC-LED drivers for reproducing advanced modulation schemes. Figure 1b shows the conventional structure, which involves the use of a high efficiency, limited-output-response Switching-Mode converter (DC–DC or AC–DC) to bias the LED, in addition to the use of a low power efficiency, very-fast-output response (i.e., wide bandwidth) RFPA to superpose the communication signal. These VLC-LED drivers focus on reproducing Single-Carrier Modulation (SCM) and Multi-Carrier Modulation (MCM) schemes that allow for providing a higher bit-rate (up to 1 Gbps). These solutions are based on classical configurations of linear Radio Frequency Power Amplifiers (RFPAs) (Class A, Class B, and Class AB) in order to obtain linearity and wide bandwidth. However, they exhibit extremely low efficiency when they manage signals with a wide ratio between average power and peak power [10–15] (i.e., a high Peak-to-Average Power Ratio, PAPR). Although the very low power efficiency only refers to power processing for communication tasks (e.g., efficiencies between 10% and 40%), this fact affects the overall efficiency of the VLC-LED driver (e.g., efficiencies below 70%). Moreover, all the proposed solutions reported to date using linear RFPAs limit the application to very short distances (up to 1 m) with very low output power for both lighting and communications tasks (up to 1 W).

As in any electronics device, a Switching-Mode Converter (either AC–DC or DC–DC [16–19]) will be in charge of generating the DC voltage level needed to supply power to any VLC system. The voltage source represented as $V_{CC}$ in the circuits shown in Figure 1 will be the output port of that

converter. No specific design constrains will distinguish this first converter to any other converter. The switching-mode DC–DC converters shown in Figure 1 are designed identically as traditional post-regulators, which are being traditionally used in LED drivers.

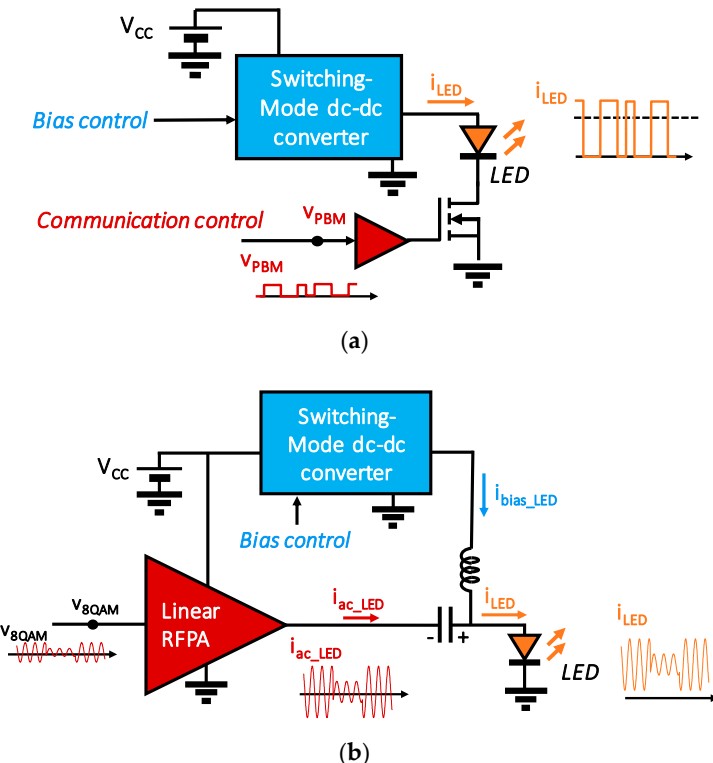

**Figure 1.** Conventional VLC-LED drivers presented to date: (**a**) VLC-LED driver based on switching the LEDs on and off, reproducing a PBM scheme; (**b**) VLC-LED driver for reproducing advanced modulation schemes, in this example, 8-Quadrature Amplitude Modulation (8-QAM).

Focusing our attention on the second subset of solutions (i.e., VLC-LED drivers for reproducing advanced modulation schemes), the low power efficiency of VLC-LED drivers is an important limitation that may be overcome so as to obtain high power efficiency and a high bit-rate. The possible solution arises from the application of traditional techniques used in the past either to improve the efficiency of linear RFPAs or to achieve linear behavior in the case of efficient Switching-Mode RFPAs. In fact, this is the main objective of the present paper: to describe the adaptation of traditional techniques used either to improve the efficiency of or linearize RFPAs to VLC. In some cases, this adaptation implies a significant change in the circuitry that defines these techniques, with major associated benefits. For all these reasons, the aim of this paper is the revision of these techniques, their adaptation for VLC, and the experimental validation of two examples: Outphasing technique and Linear-Assisted technique.

The paper is organized as follows. First, a summary of the traditional techniques used in RF to either improve the efficiency of RFPAs or to linearize the RFPAs are given in Section 2. The adaptation of these techniques to VLC are given in Section 3. The adaptation of the Outphasing technique to VLC is given in Section 4. The use of the envelope amplifiers in VLC is explained in Section 5. The adaptation of the Linear-Assisted technique is given in Section 6. Section 7 presents the circuitry design and the experimental results. Finally, Section 8 provides the conclusions of the paper.

## 2. Review of Traditional Solutions Either to Improve the Efficiency of or to Linearize RFPAs

Table 1 summarizes the traditional techniques used either to improve the efficiency of or to linearize RFPAs. On the one hand, Table 1 shows solutions for increasing the efficiency of Linear RFPAs, which are naturally inefficient when signals with very high PAPR are processed. On the other hand, there are

other kinds of techniques for providing linearity to Switching-Mode RFPAs (Class D, Class E, and Class F). Switching-Mode RFPAs are very efficient (theoretical efficiency is 100%) because the active devices (transistors) function as electronic switches instead of linear gain devices. However, they exhibit a lack of linearity because of the nonlinear relationship between the pulses that are used to drive the transistors and the output. In both cases (Linear or Switching-Mode RFPAs), these techniques are based on the use of either very-fast-output response (faster than 1 μs) Switching-Mode DC–DC converters or an arrangement of RFPAs.

**Table 1.** Traditional techniques either to improve the efficiency of or to linearize RFPAs.

|  | Linear RFPAs (Class A & B) | Switching-Mode RFPAs (Class D, E & F) |
| --- | --- | --- |
| Using Switching-Mode DC–DC converters | Envelope Tracking (ET) | Envelope Elimination and Restoration (EER) |
| Arrangement of RFPAs | Doherty | Outphasing |

In the past, very-fast-output response Switching-Mode DC–DC converters, capable of changing their output voltage at a MHz rate, have been proposed to implement techniques used to increase the efficiency of RFPAs (first row in Table 1). The most popular of these techniques are Envelope Tracking (ET) for Linear RFPAs (Figure 2a) and Envelope Elimination and Restoration (EER) for Switching-Mode RFPAs (Figure 2b) [19]. In both cases, a very-fast-output response Switching-Mode DC–DC converter has to generate the time-varying output voltage that is used as the power supply for the RFPA. The variations of this voltage are imposed by the envelope of the communication signal processed by the RFPA. This fast-output response Switching-Mode DC–DC converter is often known as an Envelope Modulator or Envelope Amplifier (EA).

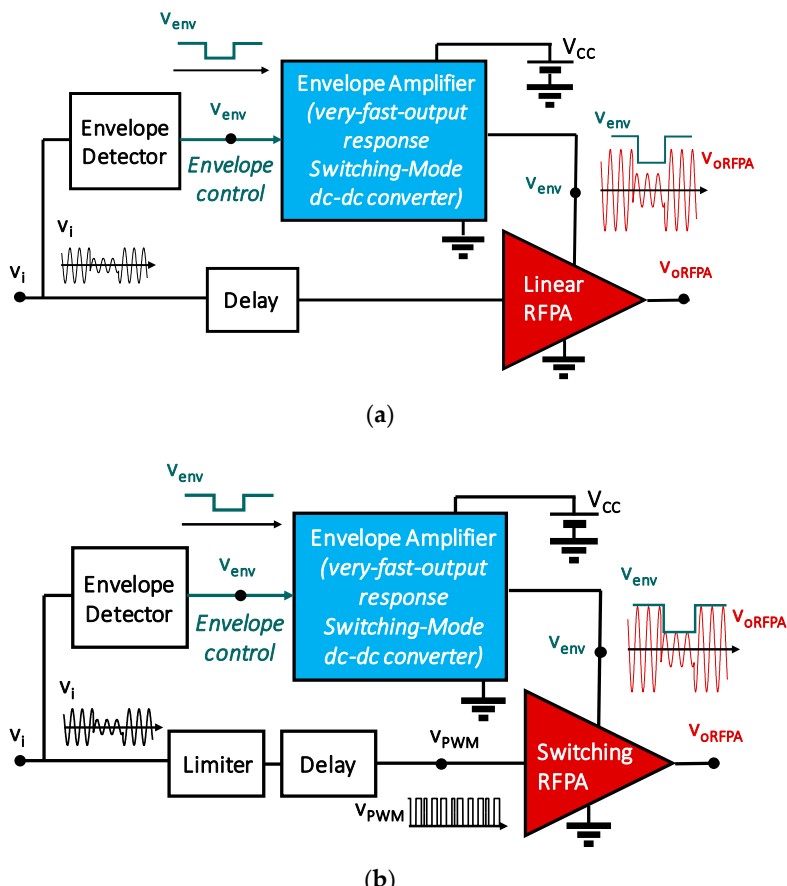

**Figure 2.** Conventional block diagram of techniques based on the use of Envelope Amplifiers (EA): (**a**) Envelope Tracking (ET) technique; (**b**) Envelope Elimination and Restauration (EER) technique.

　　　Moreover, techniques based on an arrangement of several RFPAs can achieve the desired improvements, in this case without using very-fast-output response Switching-Mode DC–DC converters: the Doherty technique for linear RFPAs and the Outphasing technique for Linear RFPAs and Switching-Mode RFPAs. W.H. Doherty originally proposed the Doherty technique in 1936 [20] (Figure 3a). In the case at hand, at least two linear RFPAs must be used. One of the RFPAs operates with the low part of the signal and is called the "carrier" RFPA. The other RFPA only works when the signal level exceeds a specific value and is called the "peak" RFPA. The realization of this technique involves the splitter of the input signal that is to be amplified and the combination of the output signal of each RFPA. The realization of the circuitry that performs these functions (i.e., the splitter and the combiner) is complex and sometimes constitutes the bottleneck of this technique. The outphasing technique was first proposed in the 1930s [21,22] and is based on the following idea: two sinusoidal signals of constant amplitude can be combined to obtain one sinusoidal signal with variable amplitude and phase. Therefore, from this statement, two Switching-Mode RFPAs working at maximum efficiency can be combined to amplify any signal (Figure 3b) by means of appropriate phase shifting between them. In this case, the design and construction of the splitter, the output network of each RFPA, and the combiner are also key points.

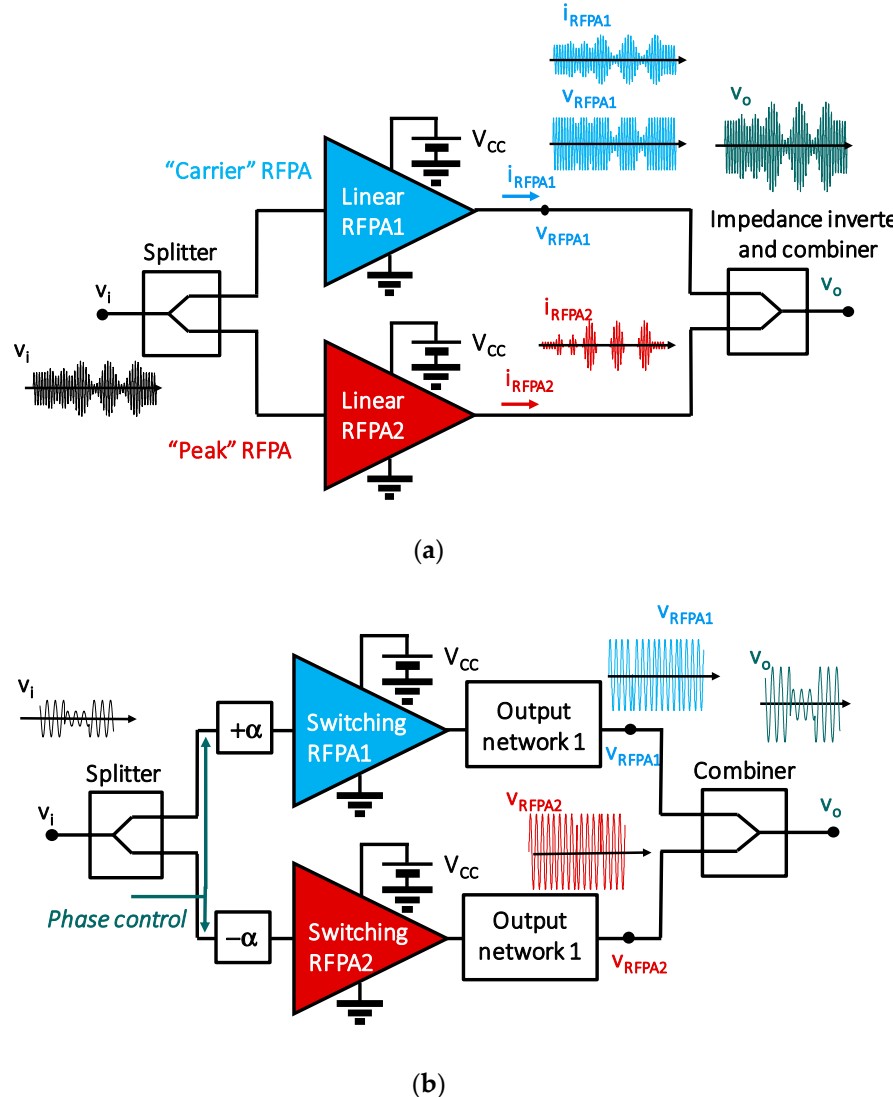

**Figure 3.** Conventional block diagram of techniques based on an arrangement of RFPAs. (**a**) Doherty technique; (**b**) Outphasing technique.

## 3. Traditional Solutions Either to Improve the Efficiency of or to Linearize RFPAs and Their Adaptation to VLC

The main idea of this section is very simple: to export and adapt any of the techniques indicated in Table 1 to the RFPA of the VLC-LED driver shown in Figure 2b. Figure 4 shows an example of the adaptation of the ET technique to VLC-LED drivers. As already stated, the ET technique requires a very-fast-output response Switching-Mode converter. This converter must include the capability of following the envelope of the signal that is to be amplified (in the example in Figure 4, an 8-QAM signal).

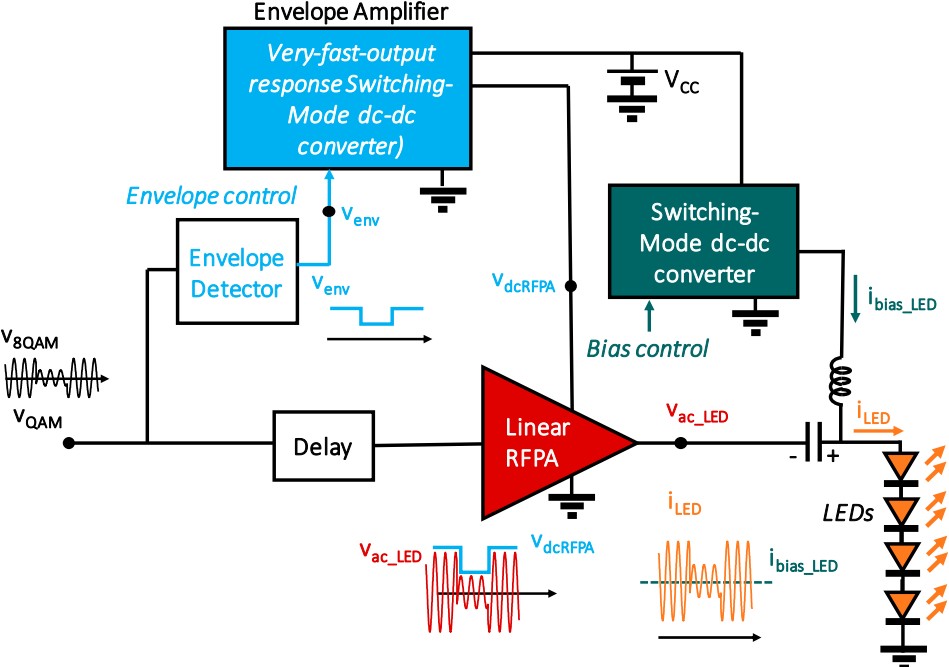

**Figure 4.** Appling the ET technique to an RFPA of a VLC-LED driver reproducing an MCM scheme (i.e., 8QAM).

However, prior to the adaptation of all the techniques in Table 1 to VLC, it is necessary to point out some aspects, which will eventually lead to the introduction of certain constraints in the real implementation:

(a)   On the one hand, the maximum bandwidth of the LED for the electrical-to-optical power conversion is a fundamental parameter that must be taken into account for VLC. A blue GaN LED in combination with a yellow phosphor LED is the preferred approach to obtain white light in Solid-State Lighting (SSL). However, this phosphor layer is optimized for color conversion, but not for achieving a very rapid response to changes in light intensity. In fact, it limits the LED bandwidth of white light to a few MHz (i.e., 3–5 MHz). Moreover, the bandwidth for blue light emission is limited to 10–20 MHz due to LED technology. On the other hand, the light intensity modulation must be fast enough to be unappreciable to the human eye for safety reasons (i.e., higher than approximately 1 kHz). The aforementioned limits define the usable frequency spectrum to modulate the amplitude and phase of the visible light intensity (VLC frequency spectrum).

(b)   The greater part of the previously defined frequency spectrum for the use of visible light (i.e., 1 kHz–20 MHz) could be used for many of the possible scenarios (in some cases, the entire VLC frequency spectrum). This fact facilitates the use of MCM schemes to achieve high bit-rates because there are no spectrum restrictions. This means it is necessary to use wideband RFPAs for VLC-drivers. Another possible solution is to use several narrowband RFPAs, whose combination

could cover the bandwidth required by the MCM scheme. However, the cost of this solution makes it unfeasible.

(c)　If very wideband RFPAs (when the BW is higher than 10% of the center frequency) operating at relatively low frequencies (due to the bandwidth of the VLC frequency spectrum) are required for VLC-LED drivers, then the adaptation to VLC of some of the techniques mentioned in Table 1 is not possible. For example, Switching-Mode RFPAs require either band-pass resonant circuits (e.g., Class D RFPAs) or transistor-integrated resonant circuits (Class E and F RFPAs) to operate properly because the square waveform generated by transistors must be filtered in order to obtain a sinusoidal carrier. This fact limits the exploitation of the VLC frequency spectrum because it is very difficult to adapt EER and Outphasing techniques using nonlinear RFPAs operating within this frequency range (e.g., from 1 kHz to 20 MHz). Moreover, in the case of the Outphasing technique, the use of selective output networks for the combiner (Figure 3b) is mandatory. The real implementation of these output networks for the VLC frequency spectrum is very complex and makes the adaptation of this technique unachievable.

(d)　In the case of the application of the Doherty technique, impedance inverters, made of discrete components, are needed. Impedance inverters are narrowband, which limit the bandwidth of the VLC-driver.

(e)　From all the above, if MCM schemes are to be reproduced for VLC, then only the ET technique can be applied to single wideband linear RFPAs. In this case, the entire free VLC frequency spectrum can be exploited.

Another option is either to give up the entire free frequency spectrum of visible light or split it up using several narrowband RFPAs. In this case, all the techniques in Table 1 can be adapted to VLC. Figure 5 shows the adaptation of the Doherty technique to VLC. As can be seen, both the "carrier" RFPA and the "peak" RFPA are highlighted. In this case, the impedance inverter is designed with discrete components (two capacitors, C, and one inductor, L) instead of the traditional quarter-wavelength transmission line, due to the frequencies used in VLC.

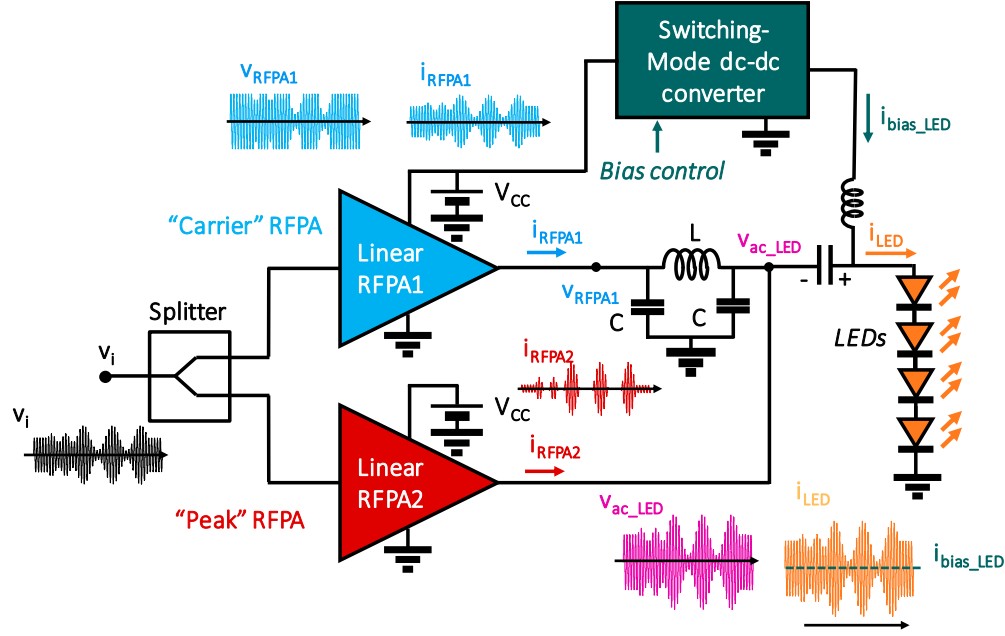

**Figure 5.** Appling the Doherty technique to a VLC-LED driver. This scheme is only valid for narrowband approaches.

## 4. Adaptation of the Outphasing Technique to VLC

The Outphasing technique can be applied using both linear and Switching-Mode RFPAs. The efficiency with linear RFPAs can be increased up to the maximum of each class (50% for Class A and 78% for B), whereas, in the case of Switching-Mode RFPAs, the maximum is theoretically 100%, making this the most suitable adaptation to VLC.

The major difficulty in applying the Outphasing technique (Figure 3b) is the design of the output combiner and the output networks, which are in charge of connecting the two RFPAs together and summing the sinusoidal signals. On the one hand, the connection of the outputs of the RFPAs is not straightforward, as the output impedance of each RFPA changes over time depending on the output voltage due to the phase shift operation, thereby leading to an undesirable influence between them. On the other hand, the combiner must be designed to prevent the differential mode connection of the load (i.e., the load must be connected to the ground of the circuitry). Traditionally, the combiner is performed by means of two quarter-wavelength transmission lines. Furthermore, the cross effect between the RFPAs due to the output combiner is especially critical when the RFPA is based on a resonant topology (i.e., Class E or Class F), in which the efficiency depends on the correct tuning of the resonant circuit. In this case, the analysis and design of the combiner is complex in order to obtain a trade-off between efficiency and bandwidth.

Another key part of the application of the Outphasing technique is the input splitter and phase control. This circuitry is in charge of generating the input signals for each RFPA, whose sum is the desired output communication signal. In the case of Switching-Mode RFPAs, this splitter and phase control can be implemented on a digital platform, as the input signals for these amplifiers are square waveforms. In the case of linear amplifiers, the splitter and phase control have to be implemented over the analog signals, thus increasing the difficulty of the implementation.

At this point, we can exploit summing the light output in order to adapt the Outphasing technique to VLC. The idea of light Outphasing is that of splitting the LED load between the two RFPAs and summing the two sinusoidal signals in their light form, as can be seen in Figure 6. In the case of Figure 6, both LED strings will generate the same DC value of light intensity due to the fact that $i_{bias\_LED1}$ equals $i_{bias\_LED2}$ (generated by a conventional Switching-Mode DC–DC converter). Furthermore, two sinusoidal signals of constant amplitude and variable phase are superposed to each string of LEDs by the actions of both Switching-Mode RFPAs. As the added quantities are two light intensities instead of two electrical signals, the need to use a combiner and connect both Switching-Mode RFPAs electrically is circumvented, thus leading to a huge simplification of the design and avoiding the influence between the two signals (with the resulting increase in power efficiency). In other words, as the outputs of each Switching-Mode RFPA are not connected together and the load is an LED string, the impedance from the point of view of the RFPAs is known and constant. Hence, the possible condition of Zero Voltage Switching (ZVS) of each RFPA does not depend on the other, thereby simplifying the overall design.

In a conventional VLC-LED driver, the LED string is connected via a bias T (Figure 1b). The bias T is used to connect the amplifier and the Switching-Mode DC–DC converter together, blocking the DC component from passing to the RFPA (using a series capacitor) and the signal components from passing to the Switching-Mode DC–DC (using an inductor). As shown in Figure 6, there is no path between the strings at high frequency, meaning that the assumption that there is no influence between the RFPAs is still correct. In the case of Class E RFPAs, due to their intrinsic resonant output filter, DC blocking is already implemented in each amplifier, so there is no need for an additional series capacitor between each RFPA and the corresponding LED string.

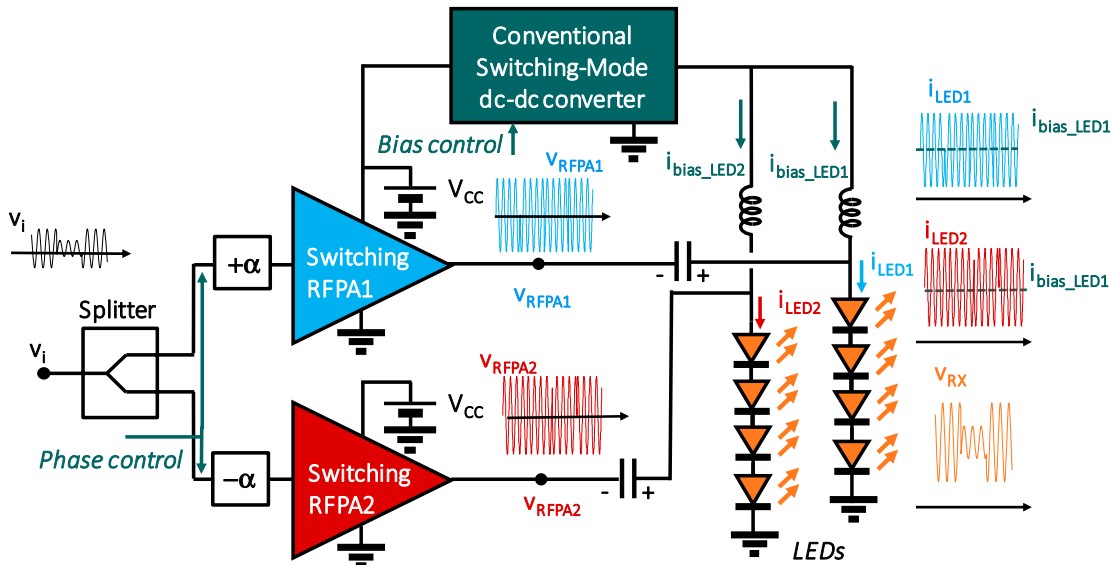

**Figure 6.** Adapting the Outphasing technique to a VLC-LED driver exploiting summing the light. This scheme is only valid for narrowband approaches.

## 5. Exploiting Envelope Amplifiers (EA) for Implementing VLC-LED Drivers

As previously stated, ET and EER techniques (Table 1) require EAs (i.e., very-fast-output response Switching-Mode DC–DC converters, Figure 2) capable of changing their output voltage at a very high frequency rate (i.e., MHz) following the envelope of the modulation scheme. The idea is now very simple: to use EAs synthesizing the overall current passing through the LED string. In this case, only one Switched-Mode DC–DC converter is also used in the final implementation of a VLC-LED driver. However, this converter is very challenging because it is in charge of generating both the DC bias current for the lighting task and the AC current level corresponding to the information to be transmitted using advanced modulation schemes (Figure 7). Therefore, the cutoff frequency corresponding to the transfer function between the control variable (the converter duty cycle) and the output voltage must be as high as the highest frequency component of the ac current that is to be used for communication tasks. In practice, this means that very-fast-output response Switching-Mode DC–DC converters must be used for this purpose.

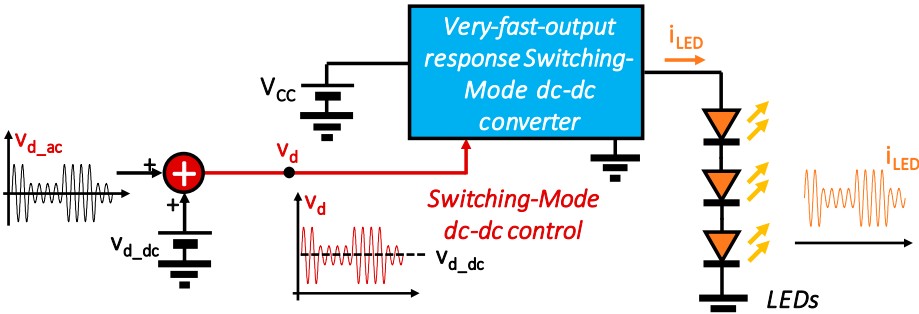

**Figure 7.** Very-fast-output Switching-Mode DC–DC converter for synthesizing the overall current passing through the LED string (VLC-LED driver). This scheme is valid for reproducing PBM, SCM and MCM schemes.

Over the years, several designs of very-fast-output response Switching-Mode DC–DC converters conceived to work as EAs have been proposed [23]. Most of these designs derive from the buck DC–DC converter, as this converter exhibits linearity between the control and output voltage operating in

Continuous Conduction Mode (CCM). The basic structure of the buck DC–DC converter (Figure 8a) presents a second-order filter at its output. Using Pulse-Width Modulation (PWM), the buck DC–DC converter can reproduce signals with frequencies up to 10–20 times its switching frequency. The challenge is now to propose circuit derivations aimed at reducing the gap between the switching frequency and the frequency of the signal to be reproduced without distortion:

(a)　Increase the order of the output filter [24] (Figure 8b). For a certain switching frequency value, the higher the filter order, the higher the rejection of the switching-related harmonics of the PWM signal at the input of the buck DC–DC converter filter. In the case of VLC-LED drivers, as the dynamic resistance of LEDs is quite constant, the conventional filter theory can be used to design a 4th- or 6th-order output filter, thus reducing the gap between the switching frequency and the frequency of the signal to be reproduced up to 4–5 times.

(b)　Use multi-input structures [25] (Figure 8c). The idea is to reduce the power of the switching-related harmonics of the PWM signal. Given that the voltage at the input of the buck DC–DC converter filter is a PWM voltage waveform, the switching-related harmonics are determined by the amplitude of the pulses. The lower the amplitude of the pulses, the lower the power of the switching-related harmonics at the input of the buck DC–DC converter filter and hence the higher the cutoff frequency of the filter for a specified switching frequency. Moreover, multi-input structures of the buck DC–DC converter mean lower switching losses due to the fact that the transistor switches between lower voltages (in the circuit in Figure 7, $S_3$ switches when $S_2$ is on and therefore withstands the voltage $V_{CC3}$-$V_{CC2}$).

(c)　Use multiphase structures (Figure 8d). The multiphase operation allows the complete elimination of the switching frequency component of PWM modulation. Although the sidebands components, which are also generated when working with a non-constant duty cycle, can only be attenuated. Moreover, this technique and the use of a higher order output filter become a very useful solution for EAs [26].

(d)　The combination of multiphase and multi-input structures is also possible [27].

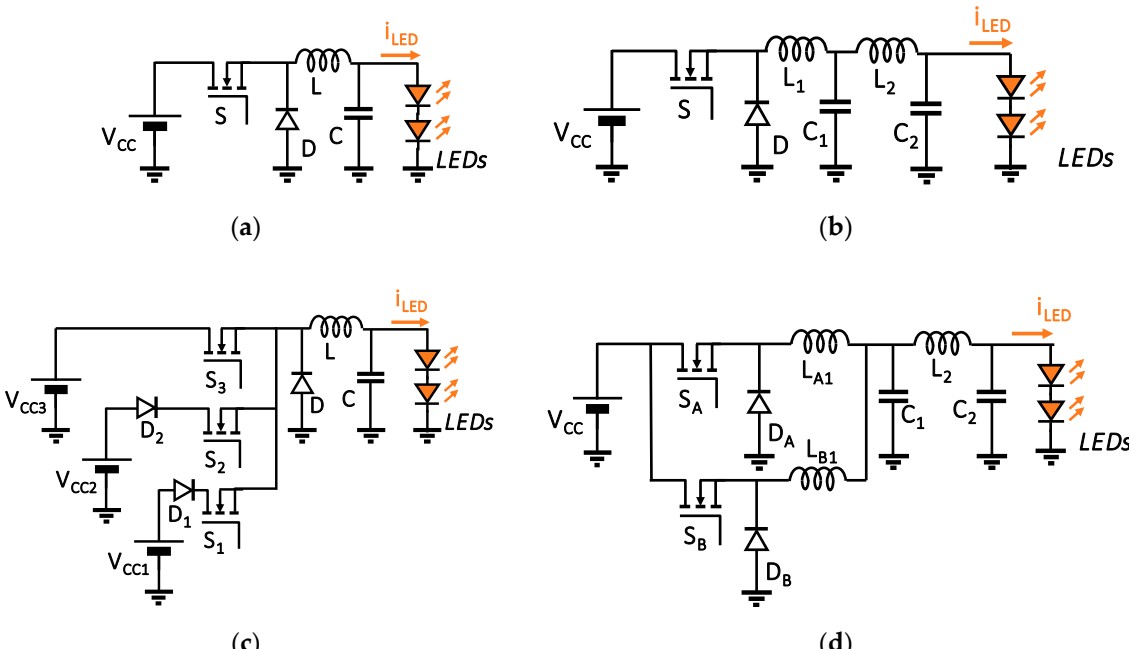

**Figure 8.** Derivations of the buck DC–DC converter. (**a**) conventional; (**b**) higher order output filter; (**c**) multi-input; (**d**) multiphase.

To increase the transient response of the Switching-Mode DC–DC converter, the combination of switching and linear stages can provide certain benefits. This solution has also been explored for

ET applications [28,29]. The proposal is based on the following idea: the greater part of the power is processed by a very-fast-output Switching-Mode DC–DC converter, while only extreme variations at the output are tracked by the linear RFPA. A good trade-off between extremely-fast-output response and high power efficiency is achieved using this solution. The two most common linear switching combinations are depicted in Figure 9. Series combination (Figure 9b) is often regarded as less efficient, as all the current through the LEDs has to pass through a dissipative linear stage. However, if the input voltage to this stage is carefully selected, it can outperform the parallel combination [29]. One of the main difficulties is to drive the transistor, S, whose source terminal is not referred to ground.

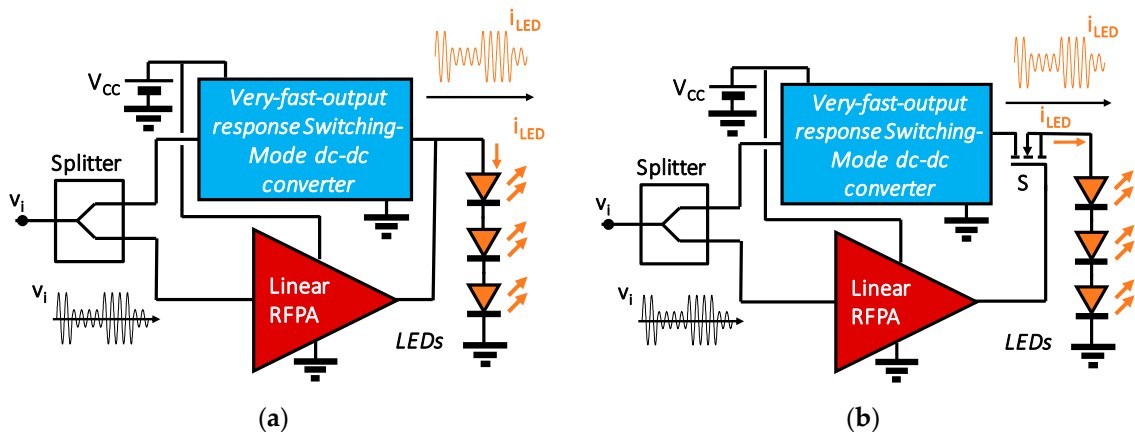

**Figure 9.** Linear-assisted technique applied to VLC-LED drivers. (**a**) parallel connection scheme; (**b**) series connection scheme.

## 6. Adaptation of the Linear-Assisted Technique to VLC

Taking the block diagram in Figure 9a as a reference, Figure 10 shows the proposed adaptation of the linear-assisted technique for VLC. The proposal follows the same principle: the high efficiency power stage comprises a Class E RFPA instead of a Switching-Mode DC–DC converter, which will be linear-assisted by a linear RFPA.

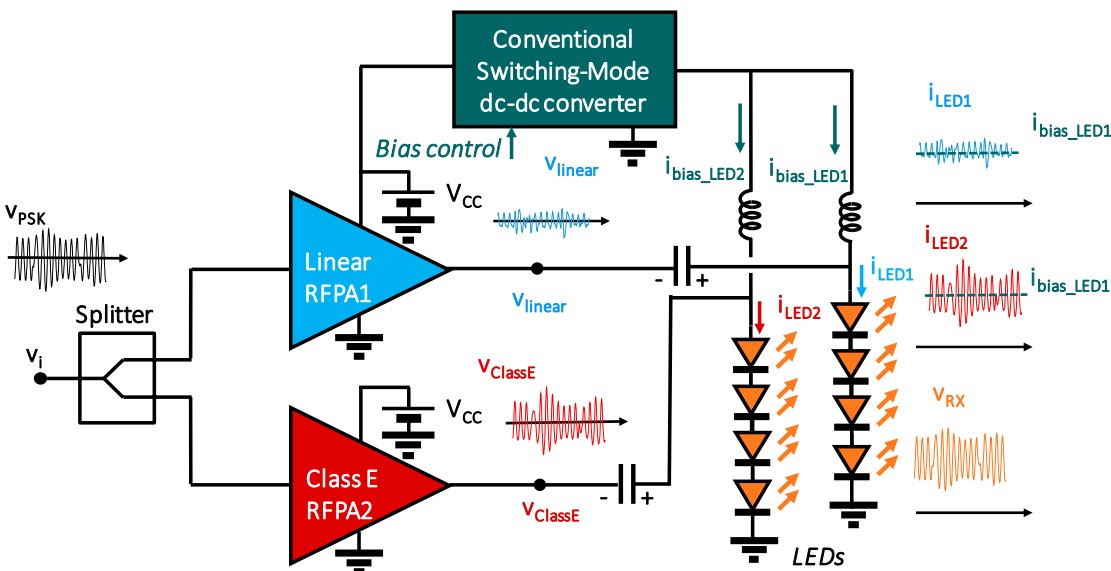

**Figure 10.** Adapting the Linear-Assisted technique to a Class E RFPA as a VLC-LED driver, exploiting summing the light. This scheme is only valid for PSK modulation schemes.

The idea of adapting the linear-assisted technique for VLC resides in exploiting the light, allowing summing both signals in terms of light, instead of electrically. This modification greatly simplifies the design. On the one hand, the design of the circuitry that combines both signals is no longer necessary, while, on the other, eliminating the electrical connection avoids the influence between the different circuitry involved. This influence is especially critical in resonant circuits such as Class E RFPAs, in which efficiency and correct behavior greatly depend on the correct tuning of the resonant circuit.

In this case (Figure 10), instead of having one string of LEDs, the load is split into two strings, each one being connected to the Class E RFPA or the linear RFPA, respectively. Both strings of LEDs are biased by a conventional Switching-Mode DC–DC converter, which controls the bias current ($i_{bias\_LED1}$ = $i_{bias\_LED2}$). By controlling the bias current, both LED strings are always working within the linear region, regardless of the temperature effect. As can be seen in Figure 10, each string of LEDs has the same bias current, thus generating the same DC light intensity in order to perform the task of illumination. Moreover, due to the connection of each string of LEDs to either the Class E RFPA or linear RFPA, its combination generates the AC light intensity related to the communication task.

With this proposed adaptation, only SCM schemes can be reproduced—in this particular case, only a Phase Shift-Keying (PSK) modulation scheme. The Class E RFPA is used to provide the great amount of power associated with the communication signal at high efficiency. The linear RFPA only supplies high slew-rate parts (phase changes) and corrects the distortion at the output.

## 7. Circuitry Design and Experimental Section

This section reports the design, building, and experimental validation of two prototypes as a proof of concept of the adaptation of the two aforementioned techniques for VLC.

### 7.1. Outphasing Technique Using Two Class E RFPAs (Exploiting Summing the Light)

The first approach is performed to adapt the Outphasing technique to VLC by exploiting summing the light. Figure 11 shows the complete circuit of the transmitter made up of: Two Class E RFPAs, two LED strings, a DC–DC converter, and an FPGA.

### 7.1.1. Circuitry Design

The Class E RFPAs were built and each one was connected to an LED string of 8 XLamp MX-3 HB-LEDs from Cree (Durham, North Carolina, USA). An external DC–DC converter is necessary to bias and control the average current through each LED string. In order to make the LED string work in the middle of its linear region, the average current is kept at 250 mA for each string. The entire process is controlled by an Artix-7 FPGA from Digilent (Pullman, WA, USA). Control of the average current is also necessary in order to avoid the temperature drift suffered by LEDs. This means that the DC–DC converter ensures that the LED always works in its linear region, regardless of the temperature. It is worth mentioning that the DC–DC converter does not required any adaptation and can be implemented using the same topologies used in normal LED drivers. The only difference is that the DC–DC converter is connected through a $L_{bias}$ = 5.6 µH biasing inductor acting as a low-pass filter. The value of the inductor is obtained as follows:

$$L_{bias} = \frac{R_L}{2\pi f_{sw}/10}, \tag{1}$$

in order to have a cutoff frequency 10 times lower than the communication signal frequency. $R_L = 17.6\,\Omega$ is the resistive equivalent of the LED string and $f_{sw}$ is the communication frequency. This enables the DC biasing current to pass to the LEDs and also provides a high impedance path for the communication signal.

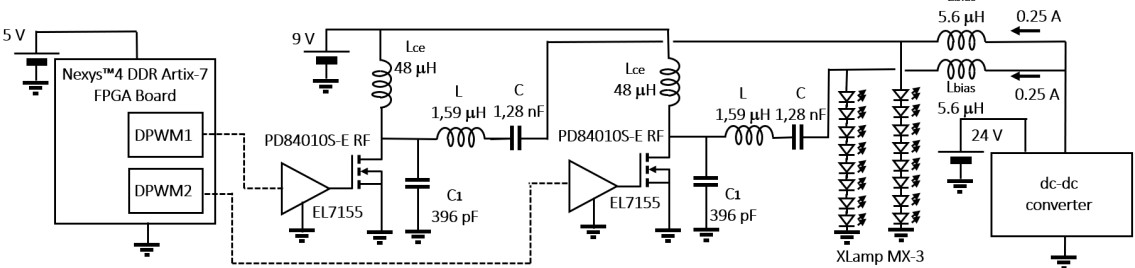

**Figure 11.** Experimental setup of the adaptation of the Outphasing technique using two Class E RFPAs.

### 7.1.2. Communication Scheme

The first part of the communication design is the selection of the communication scheme to be used. Amplitude and phase modulation are chosen to test the capability of the adaptation of the Outphasing technique. The communication scheme used is 16-QAM digital modulation with a carrier frequency of 5 MHz. The symbol period is five signal periods, reaching a bit-rate of 4 Mbps. The communication scheme defines the minimum bandwidth necessary for the amplifiers. According to [30], an approximation of the minimum bandwidth, $BW_{min}$, of the modulation is

$$BW_{min} = \frac{2}{T_s} = 2 \, MHz, \tag{2}$$

where $T_s = 1$ µs is the symbol period.

### 7.1.3. Class E RFPA Design

The circuit of the Class E RFPAs that comprises the Outphasing circuit is shown in Figure 11. Each Class E RFPA comprises a PD84010S-E RF MOSFET, a high speed EL7155 driver, a bias inductor, and an output resonant circuit. The high switching frequency needed for the communication signal requires the use of a high frequency RF MOSFET and a high speed driver. The bias inductor works as a low-pass filter to bias the MOSFET and creates a high impedance path for the communication signal. This means that no communication components flow through the power supply. The switching frequency of $f_{sw} = 5$ MHz is chosen in accordance with the communication signal. The bandwidth of the Class E RFPA depends on the quality factor, $Q$, of the output resonant filter. The higher the $Q$, the smaller the bandwidth. According to the definition of $Q$

$$Q = \frac{f_{sw}}{BW} = \frac{5 \, MHz}{2 \, MHz} = 2.5, \tag{3}$$

the value of $Q$ is 2.5 for the selected bandwidth and switching frequency. Following the design guidelines [31], the values of $L$, $C$ and $C_1$ that make up the output filter of the Class E RFPA are obtained as follows:

$$L = \frac{2.85 R_L}{2\pi f_{sw}}, \quad C = \frac{0.7124}{2\pi R_L f_{sw}}, \quad C_1 = \frac{0.219}{2\pi R_L f_{sw}}, \tag{4}$$

and the values are given in Figure 11.

### 7.1.4. Experimental Results

The main waveforms of the light-outphasing process are shown in Figure 12a. $I_{ph1}$ and $I_{ph2}$ are the currents through each LED string. The light emitted by each string is proportional to the current flowing through it. As the amplitude of the sine waves is kept constant, the amplitude of the light emitted by them is also constant. $V_{rx}$ is the light received by an optical receiver, PDA10A-EC, which receives the contribution of both strings. It can be seen in the figure that the received light is able to

reproduce amplitude changes even though the individual lights of both strings are constant. This effect is achieved due to the phase shift between the two waveforms.

The waveforms of the Class E RFPA when a phase step occurs are shown in Figure 12b. A Class E RFPA is able to reach high efficiency due to the zero voltage switching (ZVS) achieved in the main switch. The condition of ZVS is met before and after a phase shift in the Class E RFPA, as can be seen in the drain to source voltage, $V_{ds}$, but then a phase shift occurs, ZVS is lost, and hence the power efficiency decreases. Moreover, the higher the bit-rate, the more the phase shifts per second and hence the lower the efficiency. Accordingly, a trade-off between efficiency and bit-rate has to be defined in order to keep the efficiency and the bit-rate within admissible values.

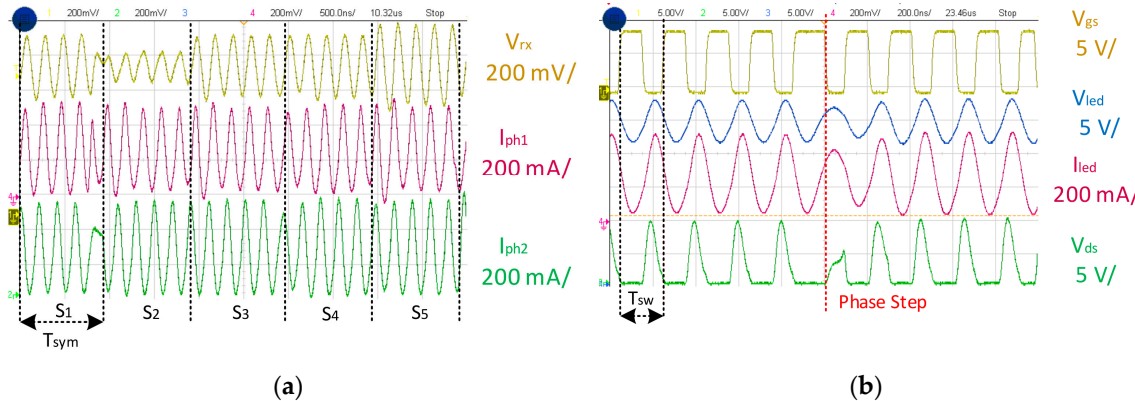

(a)  (b)

**Figure 12.** (**a**) communication waveforms of the light-Outphasing system; (**b**) main waveforms of the Class E RFPA during a phase shift.

The proposed transmitter reproduces 16-QAM digital modulation, achieving a bit-rate of up to 4 Mbps at a distance of up to 1 m. The Class E reaches an efficiency of 78% (which is the signal generation efficiency). This efficiency is measured by comparing the input DC power delivered by the 9 V biasing voltages, shown in Figure 11, and the RMS value of the communication signal in the output. The communication signal is being recorded by an oscilloscope and post processed by computer. The efficiency of the Class E is higher than the maximum efficiency reached by the Class A and B alternatives for the 16-QAM modulation. The overall efficiency reached by the whole prototype is 92%, when the signal and the lighting tasks are considered. The efficiency of the lighting task is calculated by comparing the input power of the DC–DC converter and its output power, which is the biasing power on the LED strings.

### 7.2. Linear-Assisted Technique for a Class E RFPA (Exploiting Summing the Light)

The second experimental approach is made to adapt the linear-assisted technique for a Class E RFPA. A linear-assisted Class E amplifier was suitably designed and built to provide experimental results (Figure 13). The circuitry is divided into four parts: Class E amplifier, error amplifier, linear amplifier, and external DC–DC converter. The design resides in the idea of summing the signals in terms of light, instead of electrically.

### 7.2.1. Circuitry Design

The Class E amplifier and the linear amplifier are each connected to an individual LED string comprising 8 XLamp MX-3 HB-LEDs. Both strings are biased by an external DC–DC converter which controls the average current through them. The DC–DC converter can be implemented by using a normal LED driving stage. The only modification required are the two biasing inductors $L_{bias} = 28\ \mu H$

that are used to create a biasing path for the biasing current and to avoid any communication current passing through the converter. The inductor is obtained as follows:

$$L_{bias} = \frac{R_L}{2\pi f_{sw}/10},$$

(5)

in order to have a cutoff frequency 10 times lower than the communication signal frequency. $R_L = 17.6\,\Omega$ is the resistive equivalent of the LED string and $f_{sw}$ is the communication frequency.

The Class E RFPA reproduces the communication signal, while the linear amplifier (LT1206) reproduces the error signal between the ideal communication signal and the one delivered by the Class E RFPA. The entire process is controlled by an Artix-7 FPGA.

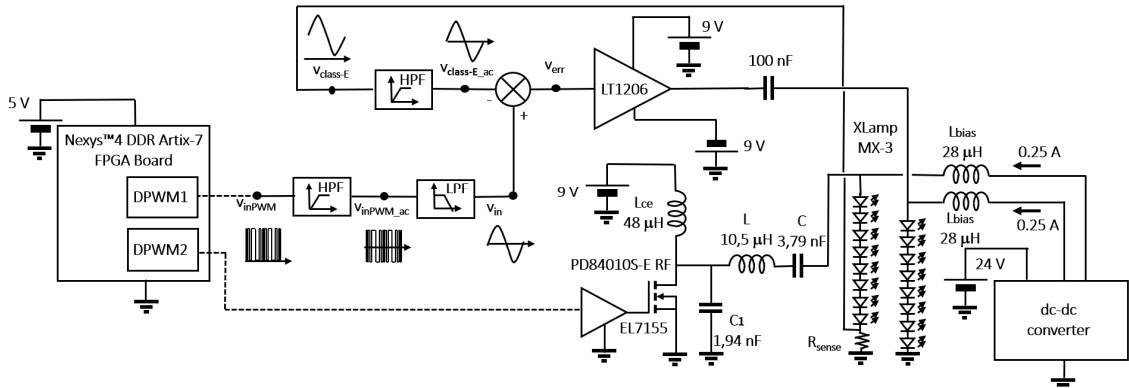

**Figure 13.** Experimental setup of the adaptation of the linear-assisted technique for a Class E RFPA.

### 7.2.2. Communication Scheme

The first part of the design consists in the selection of the communication scheme to be used. In this case, digital phase modulation (16-PSK) was chosen. The Class E RFPA reaches its maximum efficiency when the amplitude of the communication signal does not change over time. In phase modulation, the amplitude is kept constant while the phase varies over time. The chosen carrier frequency is 1 MHz, the modulation uses 16 different symbols, which encode four bits each, giving a maximum bit-rate of 0.5 Mbps. In line with what was previously done and according to [30], an approximation of the minimum bandwidth, $BW_{min}$, of the modulation is

$$BW_{min} = \frac{2}{T_s} = 250\ KHz,$$

(6)

where $T_s = 8\ \mu s$ is the symbol period.

### 7.2.3. Class E RFPA Design

Figure 13 shows the Class E RFPA circuit. It comprises the same RF PD84010S-E MOSFET and EL7155 high speed driver used previously, a bias inductor, and a resonant filter. The switching frequency is chosen according to the modulation scheme; in this case, 1 MHz is used. The resonant filter is designed according to the previously calculated modulation *BW*. According to the definition of *Q*,

$$Q = \frac{f_{sw}}{BW} = \frac{1\ MHz}{250\ kHz} = 4,$$

(7)

the necessary *Q* value to provide enough *BW* for the communications signal is 4. Following the design guidelines in [31], the values of *L*, *C*, *and* $C_1$ that make up the output filter of the Class E are obtained as follows:

$$L = \frac{3.75R_L}{2\pi f_{sw}},\ C = \frac{0.4166}{2\pi R_L f_{sw}},\ C_1 = \frac{0.2150}{2\pi R_L f_{sw}},$$

(8)

and given in Figure 13.

In order to be able to measure the error on the communication signal delivered by the Class E RFPA, a shunt resistance, $R_{sense}$, is placed in series with the LED string of the Class E RFPA. The voltage across the shunt is used in the error amplifier, generating $v_{err}$.

### 7.2.4. Linear-Assisted Circuit Design

The linear RFPA and the error amplifier are shown in Figure 13. The signal, $v_{class-E}$, passes through a 2nd-order Butterworth high-pass filter (HPF) in order to obtain only the communication signal, without the biasing contribution. The cutoff frequency is 10 kHz. Moreover, the FPGA generates the input signal modulated in PWM. In order to obtain the ideal communication signal, $v_{in}$, from the PWM, a 2nd-order Butterworth HPF and low-pass filter (LPF) are respectively needed to eliminate the DC component and the PWM harmonics. The cutout frequencies are 10 kHz and 3 MHz, respectively. Then, $v_{class-E}$ is subtracted from $v_{in}$, thereby obtaining the error signal, $v_{err}$. The error signal, $v_{err}$, is amplified by the linear RFPA, LT1206, whose output signal is delivered to a string of eight XLamp MX-3 HB-LEDs.

### 7.2.5. Experimental Results

Figure 14 shows the waveforms of the communication system reproducing 16-PSK digital modulation. The carrier frequency is 1 MHz and the achieved bit-rate is 0.5 Mbps. $v_{in}$ shows the desired and ideal communication signal that is used as a reference. $v_{class-E}$ is the signal delivered by the Class E RFPA and measured across the shunt resistance, $R_{sense}$. By subtracting $v_{class-E}$ from $v_{in}$, the error $v_{err}$ is obtained, amplified by the linear RFPA and delivered to the LED strings. An optical receiver, PDA10A-EC, receives the $v_{out}$ signal, the result of the sum in light of $v_{class-E}$ and $v_{err}$. The addition of $v_{err}$ makes $v_{out}$ closer to the ideal signal, $v_{in}$, by reducing the distortion and error introduced by the Class E RFPA, especially during phase shifts.

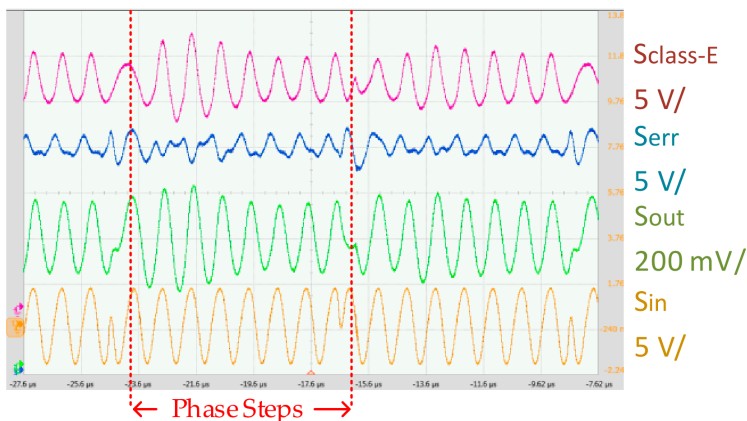

**Figure 14.** Communication waveforms of the linear-assisted transmitter.

The proposal is able to reproduce 16-PSK phase digital modulation with a carrier frequency of 1 MHz, achieving a bit-rate of 0.5 Mbps. Since the Class E amplifier delivers most of the power at a high efficiency and the linear amplifier stage only delivers the error signal, the electrical efficiency of the signal generation is 75%. This efficiency is being calculated by comparing the bias input power of both amplifiers (Class E and linear amplifier) and comparing it with the RMS value of the output communication signal. The communication signal is being recorded by an oscilloscope and post processed by computer. The signal generation efficiency is higher than the alternatives based solely on a linear amplifier. When the signal and biasing power are taken into account, the overall efficiency reached is up to 85%.

## 8. Conclusions

VLC technology is a promising alternative for alleviating the congested RF spectrum, although one of the main drawbacks of this technology is the low power efficiency of the driver when the communication capability is added. Most of the research work related to VLC focuses on increasing the bit-rate, more complex modulations, or communication schemes, among other issues, whereas little has been published on the improvement of power efficiency. This paper presents the adaptation of some RF techniques aimed at improving overall efficiency.

Two different techniques are developed and adapted to VLC by exploiting summing the light: Outphasing and linear assistance. Both techniques are based on summing two signals, which is done electrically. By using the sum of the light in both techniques, the design is greatly simplified.

On the one hand, an adaptation of the Outphasing technique is presented based on two Class E RFPAs. The proposed transmitter reproduces 16-QAM digital modulation, achieving a bit-rate of up to 4 Mbps at a distance of up to 1 m. The prototype reaches an electrical efficiency of 78% in terms of signal generation (higher than the Class A and B maximum efficiency for 16-QAM modulation) and an overall efficiency of 92% when the communication and lighting tasks are considered.

On the other hand, a linear-assisted VLC transmitter comprising a Class E amplifier and a linear amplifier is presented. The proposal is able to reproduce 16-PSK phase digital modulation with a carrier frequency of 1 MHz, achieving a bit-rate of 0.5 Mbps. Due to the fact that the Class E amplifier delivers most of the power and the linear amplifier stage only delivers the error signal, the proposal achieves an electrical efficiency of 75% in terms of signal generation (higher than the alternatives based solely on a linear amplifier) and an overall efficiency of 85% considering the signal and the biasing contribution of the DC–DC converter.

**Author Contributions:** J.R. and P.F.M. conceived and performed the experiments. D.G.A. and J.S. developed the adaptation of solutions to VLC and the design procedure. D.G.A. and D.G.L. analyzed the data and wrote the paper. V.F.R. and J.M. contributed with the definition, the design, and the evaluation of the experiments and the revision of the paper. All authors have read and agreed to the published version of the manuscript.

**Funding:** This work was funded by the Spanish Government under Project MINECO-17-DPI2016-75760-R, by the Principado de Asturias under Projects SV-PA-17-RIS3-4 and FC-GRUPIN-IDI/2018/000179, and the scholarship BP17-91, as well as by European Regional Development Fund (ERDF) grants.

**Conflicts of Interest:** The authors declare no conflict of interest.

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
