# Peer review of "Adapting Techniques to Improve Efficiency in Radio Frequency Power Amplifiers for Visible Light Communications"

_electronics, doi:10.3390/electronics9010131_

Round 1

Reviewer 1 Report

This paper presents adaptation of two techniques outphasing and linear assistance to improve the efficiency of visible light communications.

The paper is well written and interesting with some good contributions making it suitable for publication.

I have some minor suggestions to be incorporated before acceptance:

Authors may consider revising introduction by providing more literature review and highlighting contributions of this paper.

Organisation of paper may be added to the introduction.

Authors may improve quality of figure 12 and 14.

Section III can be merged with section 2.

Reviewer 2 Report

The reviewer has the following comments:

You may want to clarify the type of switch-mode power supply shown in fig. 1 and fig. 2 – flyback, buck, buck-boost, etc. in the introduction section as it is the major object of your research. Is it necessary to be transformerless only or in some cases transformer based converter could be necessary? Could you formulate recommendations about its topology selection?

Your research will be better presented if you quantify expressions such as “high bit-rates” (line 58), “very efficient” (line 108), “very-fast-output response” (line 133), “very wideband” (line 173), etc. using numerical values. Such expressions without numerical values are ambiguous and do not have scientific soundness. In addition, the literature review could be improved.

Considering that the manuscript is an article, not a review, do you really need to cite works from the 1930s? The introduction part looks more like a history notice rather than literature research.

It is a good style the introduction section to finish with a paragraph clearly showing the aim and the novelty of the presented research. For example, it is correctly shown that the linear topologies have much lower efficiency than switch-mode DC-DC converters, which is applied in the proposed VLC-LED. As this is a well-known fact, the novelty in your research must be shown better.

Sections 2 and 3 “Traditional solutions …. “ can be united and improved. The entire structure of the paper looks messy and needs some reorganization. For example, the literature review continues in section 5 (line 263). The style should be improved as in some places sounds like a fairy tale.

A better analysis of all circuits could be offered. The switch-mode DC-DC converter shown in fig. 8 is a Buck converter with its varieties, i.e. as “conventional” can be described every Boost, Full-bridge, Half-bridge, etc. converter. As this converter is an important part of your research, you could have offered better analysis. For example, you could show the numerical values for the switching frequency, and its dependency on the signal frequency supported by diagrams based on your data. Considering the required frequencies, what type of elements for the L-C filter and the semiconductors have you used? Would you clarify the multilevel filter topology (line 271 – line 276) using your data? The same questions about the multi-input and multiphase (interleaved) circuit. Having supported the circuit analysis with data obtained from your research, you would show explicitly your contribution to the field. Currently, your circuit analysis looks like just borrowed from a textbook without your personal contribution.

You may want to separate the experimental set-up from the circuit design. Both could be significantly improved. The readers would be interested to see more about the elements value calculations, which are shown in fig. 12 and fig. 13. Have you used the standard design procedure for the DC-DC converter or due to the specific application some modifications are necessary? Your paper would gain significant interest and hence citations if you manage to offer a design clarification of the circuits given in fig 11 and fig. 13.

Considering the frequency ranges what are the conditions ZVS to be achieved? As your aim is efficiency improvement, this should be shown in the experimental set-up with the necessary measurements, graphics, etc. Currently, in the conclusion section, an efficiency of 78% and overall efficiency of 92% (line 458 – line 468) for your circuit are given, but they just appear out of nowhere. The presentation of your experimental verification would be significantly improved if you show how exactly these results are accomplished.

The reviewer would assume that your research has significant potential, but the paper needs a major revision.

Round 2

Reviewer 2 Report

Thank you for your effort.

The reviewer agrees with the provided answers. As the paper has been  improved the reviewer would suggest that it can be published in the present form.

Best regard.